

# Impact of reflected shortwave anisotropy on satellite radiometer measurements of the Earth's energy imbalance

Thomas Hocking[1,2], Linda Megner[1,2], Maria Hakuba[3], and Thorsten Mauritsen[1,2]

[1]Department of Meteorology, Stockholm University, Sweden
[2]Bolin Centre for Climate Research, Stockholm University, Sweden
[3]Jet Propulsion Laboratory, California Institute of Technology, USA

**Correspondence:** Thomas Hocking (thomas.hocking@misu.su.se)

**Abstract.** The Earth's energy imbalance is the difference between incoming solar radiation and outgoing reflected and emitted radiation from the Earth, and quantifies the current ongoing accumulation of energy in the Earth's climate system. There are indications that the imbalance is growing, and it is important to be able to measure and monitor this quantity to better constrain future changes. The reflected shortwave component of the outgoing radiation depends on surface and atmospheric properties, which leads to strong directional variations associated with the angular geometry relative to the incoming sunlight. This means that a reflected shortwave radiance measurement at a specific point in space and time may differ by an order of magnitude between an assumed isotropic case and a case with more realistic anisotropic reflection. The effect of this anisotropy on global average measurements from wide-field-of-view radiometers has been the topic of some investigation in the past, and results from an earlier study suggest that this effect could potentially lead to substantial systematic biases in the context of the global mean reflected shortwave radiation. Here we simulate wide-field-of-view instruments on satellites in polar, sun-synchronous and precessing orbits, as well as constellations of these types of satellite orbits, with both Lambertian (isotropic) and anisotropic shortwave reflection. We find that both the estimated global annual mean and the estimated interannual trend only exhibit limited sensitivity to whether Lambertian or anisotropic reflection is assumed. With anisotropic reflection, the estimated global annual mean root-mean-square sampling error is at most $0.11$ $\mathrm{Wm}^{-2}$ provided that at least two complementary satellites are used, compared with at most $0.09$ $\mathrm{Wm}^{-2}$ in the case of Lambertian reflection. The magnitude of the difference in the estimated interannual trend is at most $0.07$ $\mathrm{Wm}^{-2}$ per decade, and typically only $\sim 0.01$ $\mathrm{Wm}^{-2}$ per decade. Analysis of the angular sampling of these satellites reveals that the anisotropic reflection requires sufficient sampling of viewing zenith angle and relative azimuth angle, in addition to the solar zenith angle. However, we conclude that it is possible to choose satellite orbits so that the sampling error is not substantially affected by reflected shortwave anisotropy.



## 1 Introduction

The Earth's energy imbalance (EEI) quantifies the difference between incoming and outgoing radiation for the Earth system relative to surrounding space. The shortwave EEI component is determined by the incoming and outgoing reflected sunlight, while the longwave component is given by thermal emission from the Earth. In a steady state of the climate system, this difference would be a small net loss to space that matches the geothermal flux of $0.09 \ \mathrm{Wm}^{-2}$ (Davies and Davies, 2010).

In contrast, the IPCC AR6 puts the current best estimate of the total EEI at 0.79 (0.52 to 1.06) $\mathrm{Wm}^{-2}$ net incoming for the period of 2006-2018 (Forster et al., 2021). This is an increase compared with values for the late 20th century, and observations since the start of the 21st century indicate that the EEI is increasing steadily, e.g. at a rate of $0.42\pm0.23 \ \mathrm{Wm}^{-2}\mathrm{decade}^{-1}$ for the period of 2000-2020 (Loeb et al., 2022). This increase leads to faster accumulation of energy in the Earth system, which has unequivocal impacts on the climate of our planet, such as an increased frequency and intensity of extreme events

(e.g. Seneviratne et al., 2021). Because of these impacts, it is obviously desirable to monitor the accumulation of energy, and one way is to track interior energy increases in different parts of the Earth system in the form of heat inventories (e.g. von Schuckmann et al., 2023; Hakuba et al., 2024a). However, since the increases are a result of accumulated energy rather than instantaneous power, it takes time for EEI-related changes and trends to stand out from short-term variability and measurement noise. It is challenging to quantify the EEI on sub-decadal time scales in this way due to e.g. insufficient spatio-temporal

coverage of ocean heat content measurements (Meyssignac et al., 2019; Hakuba et al., 2021). By directly observing the EEI instead of accumulated quantities, for instance with satellites, detection would be possible on shorter time scales and could e.g. allow us to distinguish between global emission scenarios many years earlier than with temperature-based measurements (Meyssignac et al., 2023).

Historically, satellites have been used to measure the Earth radiation budget (ERB) components since the 1950s (House

et al., 1986). Explorer 7 was launched in 1959, and provided ERB data for seven months. This was followed by several other individual satellites. In the 1970s, the first satellites were launched that would provide ERB consecutive data for multiple years. The three satellites ERBS, NOAA-9 and NOAA-10 were launched in the 1980s as part of the multi-satellite mission Earth Radiation Budget Experiment (ERBE) (Barkstrom, 1984). Many of these early missions used wide-field-of-view radiometers, but these instruments suffered from e.g. thermal issues caused by direct solar illumination, which meant that scanning radiometers

were used in later missions, such as the current Clouds and Earth's Radiant Energy System (CERES) instruments on the Aqua, Terra, NOAA-20 and Suomi-NPP satellites (Wielicki et al., 1996; Wong et al., 2018). When it comes to future measurements, there are satellite missions for ERB and EEI measurements at various stages of planning, such as the Libera mission that aims to ensure continuity of the CERES record while also introducing expanded measurement capabilities (Hakuba et al., 2024b). Other alternative or complementary missions are also under development at various stages of maturity, and some of these

mission concepts indicate that there is renewed interest in wide-field-of-view measurements and related techniques (Schifano et al., 2020; Hakuba et al., 2019; Hocking et al., 2024).

A significant concern regarding these wide-field-of-view instruments is their performance and the effect on the retrieval of the global mean under anisotropic conditions. In this context, anisotropy refers to the angular inhomogeneity of outgoing





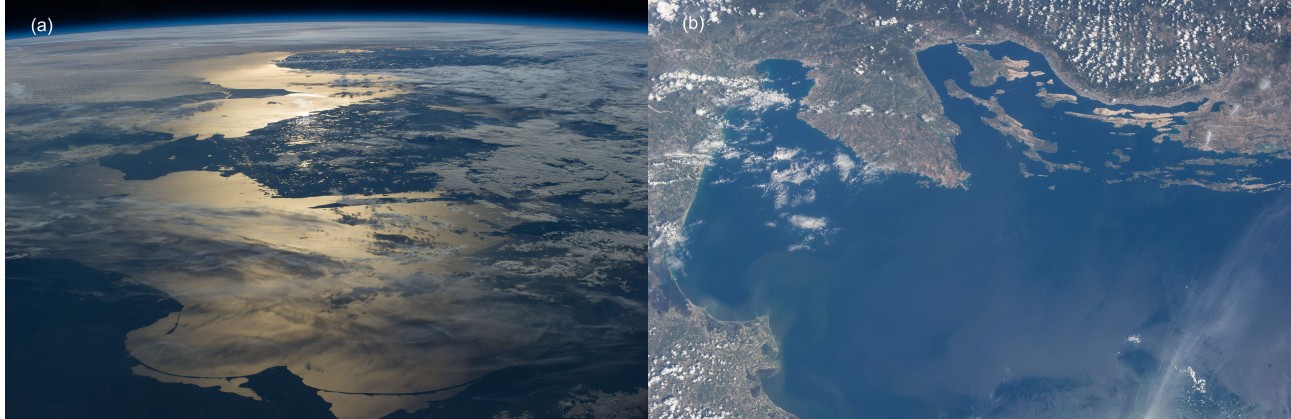

**Figure 1.** Views from the International Space Station that illustrate anisotropic shortwave reflection. (a): View over Kattegatt and the Baltic Sea showing strong ocean reflection in certain directions. (b): View over the Adriatic Sea showing weaker reflection within the image. Photos: NASA (left) and Earth Science and Remote Sensing Unit, Lyndon B. Johnson Space Center (right). Both public domain, via Wikimedia Commons.

radiation from a given surface element. A practical example is shown in Fig. 1a, where the ocean surface acts as a mirror and

55 reflects solar radiation very strongly at certain angles, in contrast to the land and cloud surfaces that reflect very little radiation in the same directions. This demonstrates that anisotropy has a strong impact on individual measurements, which is something that is important for high-resolution analysis of the radiation budget at the top of the atmosphere. However, it is less obvious how this should affect low-resolution analysis, such as estimates of the annual global mean. On the one hand, total energy is conserved and as such the global mean should not be affected if it is sampled everywhere all the time, but on the other hand,

the sampling of a given satellite instrument may be biased in terms of the observation angles and may thus result in an incorrect annual global mean.

An earlier study by Gristey et al. (2017) indicates that the effect of anisotropy could be substantial in the daily global mean, which is something that is important to consider for investigations on annual time scales. The authors considered a hypothetical constellation of a total of 36 satellites in six different orbital planes, where each satellite was equipped with

65 a nadir-looking wide-field-of-view radiometer. The satellites had an orbital altitude of 780 km and an orbital inclination of 86.4°. Radiometer measurements were then simulated for one day with both an isotropic and an anisotropic assumption for the shortwave radiation, where the anisotropy relied on angular distribution models from CERES (Loeb et al., 2003b). In terms of the global mean, Gristey et al. (2017) found that the constellation had a shortwave bias of $0.16 \pm 0.45 \ \mathrm{Wm^{-2}}$ in the isotropic case, and that this bias increased by $0.56 \ \mathrm{Wm^{-2}}$ in the anisotropic case. Sensitivity studies further indicated that the increased

bias was mostly due to the viewing geometry itself, rather than the scene type for the angular dependence model. Investigations by Wu et al. (2022) and Huang et al. (2023) in the somewhat different context of Moon-based measurements did not address the potential systematic bias in the global mean, but identified the relative azimuth angle as an important parameter in assessing anisotropic radiation.



In this study, we will perform a similar analysis to that in Gristey et al. (2017), but we investigate expected measurements from small constellations over multiple years, rather than the measurements from a larger constellation over one day considered by Gristey et al. (2017). We build on our previous work for the proposed Earth Climate Observatory (ECO) satellite mission, the goal of which is to measure the EEI by differential measurements with wide-field-of-view radiometers facing the Earth and the Sun (Schifano et al., 2020; Hocking et al., 2024). In this case, there is limited focus on high resolution in space or time. Instead, the desire is to achieve as good a measurement as possible of the global mean EEI itself, nominally on an annual time scale. As part of previous work, we simulated and investigated the sampling error for different orbits with assumed Lambertian shortwave radiation (Hocking et al., 2024). Contextual information and methodology descriptions from that study are partly repeated in the current document, but the full details can be found in the original publication. In the current study, we extend the simulations to include anisotropic shortwave reflection in order to assess the impact of anisotropy and its implications for the global mean sampling error.

## 2 Methods

The following sections present the methods used to simulate what an idealised satellite would measure, in situations with either Lambertian or anisotropic shortwave reflection. The methods are largely identical to those used in Hocking et al. (2024), with some extensions and adjustments for the investigation of shortwave anisotropy. Here, we repeat the main points of the relevant methods from Hocking et al. (2024), but refer to that study for further information. The new aspects of the method are described in more detail.

### 2.1 Radiation at the top of the atmosphere

We use data from the CERES SYN1deg product (NASA/LARC/SD/ASDC, 2017) as our reference truth, in order to have fluxes that are representative of the real geographical and temporal distribution. We investigate the period 2001-2005. The bias of approximately 5 $\mathrm{Wm}^{-2}$ in the EEI in this data product relative to current best estimates (Loeb et al., 2018) does not affect our study, as we consider the deviation from our reference truth rather than the absolute values.

CERES SYN1deg provides hourly fluxes at the top of the atmosphere (TOA) on a $1°\times1°$ latitude-longitude grid. Specifically, we use the all-sky incoming shortwave, outgoing shortwave and outgoing longwave radiation fields, and for simplicity assume that the Earth emits radiation as a spherical shell, with the TOA located at zero altitude. For the incoming shortwave radiation, we only consider radiation that hits the surface of the Earth; light that passes through the atmosphere without hitting the surface, for instance during twilight, is thus not considered in the current study. The outgoing longwave radiation is treated as Lambertian, i.e. isotropic. In principle, this radiation is not entirely isotropic, but a Lambertian approximation is still reasonable since the anisotropic effect is significantly less pronounced than for shortwave radiation (e.g. Suttles et al. (1988, 1989)). The outgoing shortwave radiation is our main focus, and it is computed using both a Lambertian and an anisotropic implementation. Details on the Lambertian implementation can be found in Hocking et al. (2024), whereas the anisotropic treatment is introduced below.





## 2.2 Emulating anisotropic shortwave reflection with angular distribution models

As illustrated in Fig. 1, shortwave radiation can have a strong dependence on the angular geometry between a given surface element and an observing satellite. Our input data from CERES are given as total outgoing fluxes for each grid element, so we need to translate these fluxes into directional radiances. To do so, we use angular distribution models (ADMs), which are ordinarily used to translate directional radiances into total fluxes (Gristey et al., 2021). Before describing how we apply ADMs in more detail, we first clarify the geometry involved.

### 2.2.1 Angular geometry: zenith angles and relative azimuth angle

The solar zenith angle determines the apparent vertical position of the Sun, as seen from a surface element: at 0° the Sun is in zenith, while at 90° it is at the horizon. The viewing zenith angle likewise describes the vertical position of the observer, which in our case is a satellite, relative to a surface element. The Sun and the satellite also each have an azimuth angle that specifies the position in the surface local horizontal plane. This angle is conventionally defined relative to local north, but the important quantity for the ADMs is the relative azimuth angle between the Sun and the satellite: it varies between 0° when the satellite is directly "opposite" the Sun, and 180° when the satellite is exactly aligned with the Sun. Thus the Sun-surface-satellite geometry is completely described by the three angles: solar zenith angle, viewing zenith angle and relative azimuth angle.

### 2.2.2 Angular distribution models

An ADM describes the outgoing radiance field from a surface element in different directions, relative to an idealised case with so-called Lambertian, i.e. isotropic, outgoing radiation. For this Lambertian reference case, the outgoing radiance $I_L$ along a zenith angle $\theta$ is

$$I_L(\theta) = \frac{M \cos(\theta)}{\pi}, \tag{1}$$

where $M$ is the radiant exitance of the surface element in question. If a surface exhibits perfectly Lambertian emission, the full outgoing radiation from the surface can then be retrieved from a single radiance measurement by simply inverting Eq. 1. By contrast, a more realistic case with anisotropic emission does not allow this simple retrieval, because the scattering properties of the surface are not known a priori. The ADM describes these scattering properties as a collection of anisotropic factors $R$ that are defined as the ratio of the actual radiance $I$ to the Lambertian radiance. For reflected shortwave radiation, $R$ is usually presented as a function of solar zenith angle $\theta_0$, viewing zenith angle $\theta$ and relative azimuth angle $\phi$ (Suttles et al., 1988):

$$R(\theta_0, \theta, \phi) = \frac{I(\theta_0, \theta, \phi)}{I_L(\theta)}, \tag{2}$$

where the anisotropic factors are normalised so that the integrated sum for any solar zenith angle is unity, i.e. they represent a spatial redistribution of the total flux, with the overall magnitude conserved. With an anisotropy factor from an appropriate ADM, even an anisotropic radiance measurement can then be translated into a corresponding Lambertian radiance and, by extension, into a total outgoing flux. In our case, we use the ADMs in the opposite sense: starting from a known total outgoing flux, we can compute the radiance in any direction as $I = R \cdot I_L$.



Of course, an ADM is necessarily a simplification. These ADMs are typically constructed empirically from long measurement time series, which over time cover all the necessary combinations of angles, and are divided into distinct "scenes" according to anisotropic behaviour. The criteria for what defines a scene may vary, depending on the desired complexity of a given collection of ADMs (e.g. Suttles et al., 1988; Su et al., 2015).

### 2.2.3 ERBE angular distribution models

For this study, we use ADMs that were developed for ERBE (Suttles et al., 1988). Our choice to use these ADMs was based on a desire for a reasonable representation of anisotropic reflection that was also feasible to implement in our framework. While ADMs from CERES might seem like a natural choice in combination with the CERES fluxes, the source of ADMs can in principle be chosen separately from the fluxes. For reference, the CERES ADMs require external information for multiple meteorological variables as part of scene identification, which increases the complexity of the implementation (Loeb et al., 2003b). By contrast, the ERBE ADMs only require a limited number of additional inputs to determine the scenes, which greatly facilitated the practical execution of this study. In terms of the resulting representation of anisotropy with ERBE ADMs, the anisotropy effect may be exaggerated because of a systematic increase in albedo with viewing zenith angle compared with improved ADMs from e.g. CERES (Loeb et al., 2003a; Gristey et al., 2021). On the other hand, the ERBE ADMs may underestimate the anisotropy effect because they are less complex, and do not capture differences in anisotropy related to e.g. cloud optical depth and cloud phase (Gristey et al., 2021). Overall, we consider the chosen ADMs to provide a reasonable representation of anisotropic reflection.

The ERBE ADMs use 12 different scene types, based on combinations of a time-constant surface type (Fig. 2a) and a time-varying cloud fraction (Fig. 2b). Figure 2c shows an example snapshot of the TOA reflected shortwave field that we use as input to our method to emulate anisotropic shortwave reflection. The resulting anisotropy profiles can look quite different depending on the scene type, as illustrated in Fig. 3. For example, the "Clear Ocean" scene scatters more in the forward direction while "Clear Land" exhibits more uniform scattering, similar to the situation shown in Fig. 1. The full scene classification is detailed in Table 1.

On a technical note, the time resolution of our input data requires some caution when using these data in combination with ADMs, to avoid a non-physical loss of energy in our framework. Because the input radiation fields only have hourly resolution, grid elements that are illuminated for only part of an hour will effectively appear as being illuminated for the full hour. This is relevant near dusk and dawn, when the true illumination comes from solar radiation near the horizon. However, a straightforward implementation that includes the position of the Sun may determine that the Sun is in fact slightly below the horizon, and as such set the illumination to zero. This would result in an overall loss of energy relative to the original hourly input data, and would complicate comparisons with the reference truth. In order to make sure that total energy is conserved in these cases, we therefore extend the ERBE ADMs, which are otherwise only given for solar zenith angles up to 90° i.e. at the horizon.



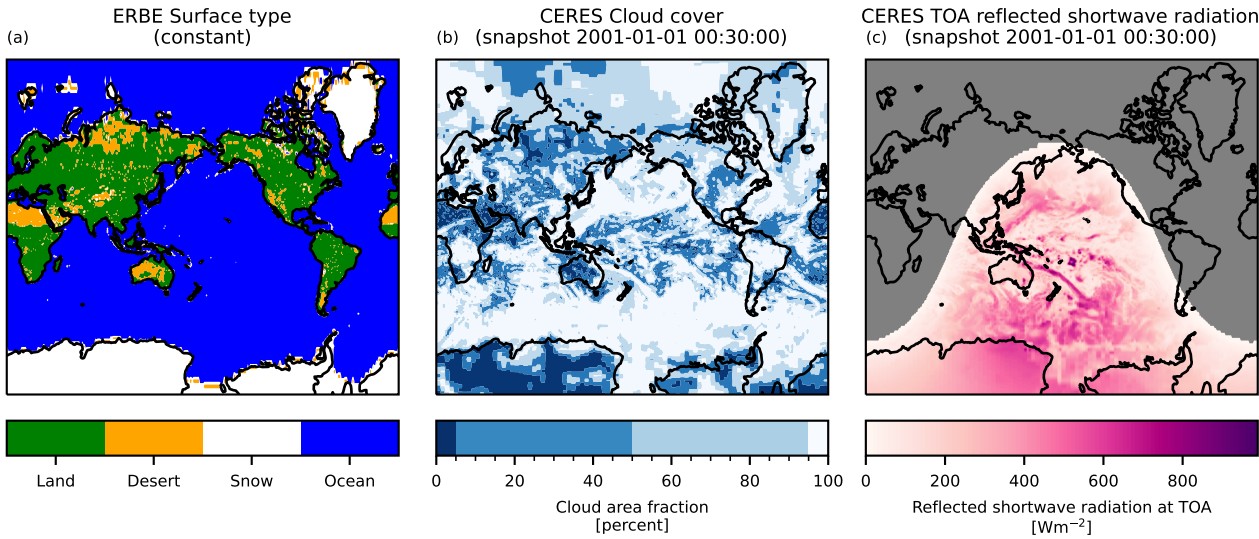

**Figure 2.** In our framework, the reflected shortwave radiances from each grid element are determined using three input components: the surface type (panel a) and the cloud fraction (panel b) together determine the appropriate angular distribution (Sect. 2.2.3), while the TOA field from CERES SYN (panel c) determines the magnitude of the total flux.

In our extended ADMs, the anisotropy factors corresponding to solar zenith angles of 90° are duplicated for angles greater than 90°, which essentially provides a small tolerance for when the Sun is slightly below the horizon. This extension does not affect the results for grid cells and time steps where there is no illumination at all because the ADM conserves the total magnitude of the radiation, which in this case is zero. It is also worth pointing out that this issue of energy loss is only an artefact caused by the discrete nature of the input data, and not something that would affect real measurements.

In principle, the choice of which anisotropy factors to duplicate for the ADM extension does come with a systematic bias in the angular distribution of the measured shortwave radiation, but in practice, this choice does not substantially affect our results. To assess the potential impact of this bias, we performed comparative one-year simulations with alternative ADM extensions, choosing to duplicate anisotropy factors for slightly smaller solar zenith angles, i.e. slightly higher solar positions. We determined that the resulting single-satellite global means differed by up to a few tenths of a $\mathrm{Wm^{-2}}$ for sun-synchronous satellites, and at most $0.01\ \mathrm{Wm^{-2}}$ for other satellites (the orbits are described in Sect. 2.3), and concluded that the overall effect on the results on this study was small.





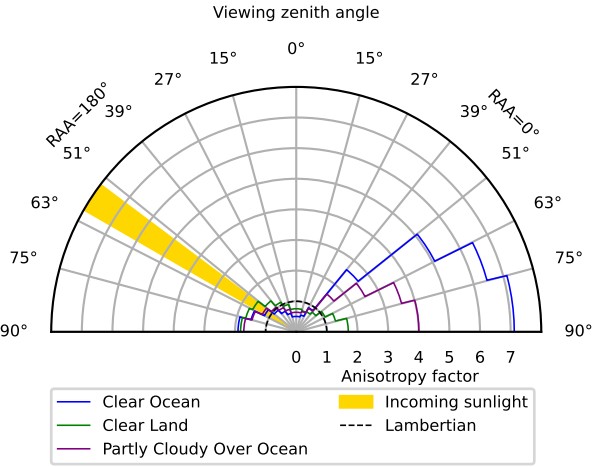

**Figure 3.** Illustration of selected anisotropy factors for three scene types from ERBE ADMs. The two-dimensional side view is a cross section at the principal plane, i.e. along 0°/180° relative azimuth angle (RAA) and aligned with the incoming solar radiation. The displayed values are for incoming sunlight with solar zenith angles between 53.13° and 60.00°. The viewing zenith angle grid lines correspond to the bins in the ERBE ADMs. The anisotropy factors are normalised such that the integral over solid angle is unity for each solar zenith angle, which ensures that the total radiation is conserved. Even though the cross-sectional areas in this two-dimensional figure do not match that of the Lambertian case, these differences vanish once the out-of-plane contributions from other relative azimuth angles are included.





| Cloud fraction | Surface type | Scene index |
|---|---|---|
| 0%-5% (Clear sky) | Ocean | 1 |
| | Land | 2 |
| | Snow | 3 |
| | Desert | 4 |
| | Mixed | 5* |
| 5%-50% (Partly cloudy) | Ocean | 6 |
| | Land/Desert/Snow | 7 |
| | Mixed | 8* |
| 50%-95% (Mostly cloudy) | Ocean | 9 |
| | Land/Desert/Snow | 10 |
| | Mixed | 11* |
| 95%-100% (Overcast) | Any | 12 |

**Table 1.** ERBE ADM scene classification, adapted from Suttles et al. (1988). *These mixed scene types were not used in the current study since a priori knowledge allows us to process each individual grid cell separately, with a single well-defined surface type.

### 2.2.4 Resulting reflected shortwave radiation

The set of ERBE ADMs can then be used in combination with the reflected shortwave irradiance field at the TOA to generate the observed reflected shortwave radiation at the satellite level. A sample snapshot of the CERES SYN1DEG TOA field is shown in Fig. 2c, while the resulting satellite-level field is illustrated in Fig. 4. Lambertian and anisotropic reflection both lead to smoothing of the small-scale TOA features, and the absolute fields are difficult to distinguish at first glance, but the difference map (Fig. 4c) clearly shows that there is a systematic change between the two types of reflection. This systematic change shows that it is important to consider how potential satellites would sample the large parameter space.

### 2.3 Satellite orbits

We consider idealised circular orbits at an altitude of 700 km with four different inclinations: 73°, 82°, 90° and 98°, as in Hocking et al. (2024). These orbits are illustrated in Fig. 5. For each inclination, we simulate 12 satellites whose orbital planes are evenly spaced around the Earth. These planes are initialised according to their right ascension of the ascending node (RAAN), which is equivalent to the mean local time. The 98° orbit is a sun-synchronous orbit, which always observes the Earth at the same local solar time by definition. The 90° orbit stays fixed in an Earth-centred, non-rotating frame, and therefore gradually drifts in observed local time as the Earth moves along its orbit around the Sun, and samples all local times twice per year. The 82° and 73° orbits precess, and therefore drift faster in local time than the 90° orbit, sampling all local times four and



six times per year, respectively. The inclination of a satellite also determines the maximum latitude that the satellite reaches, which affects the geographical sampling.

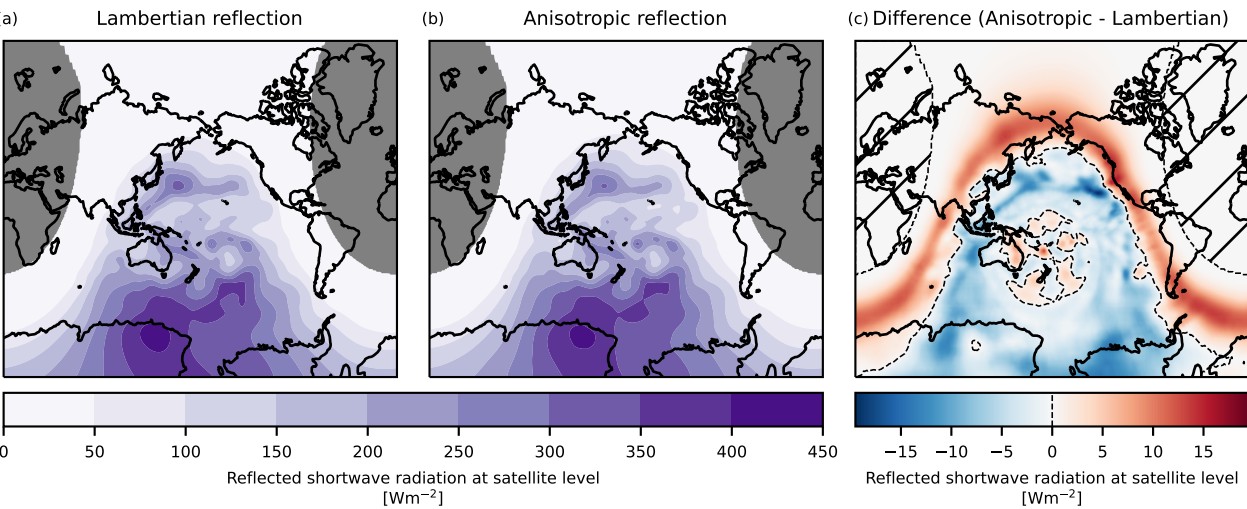

**Figure 4.** Difference between synthetic satellite-measured reflected shortwave radiation at a satellite altitude of 700 km using Lambertian or anisotropic reflection. Panels a and b are the results of reflected shortwave satellite measurements all over the Earth, i.e. effectively convolutions of the shortwave field in Figure 2c with Lambertian or anisotropic shortwave measurement kernels, respectively. The synthetic measurement satellites are artificially positioned in grid cells every 1° in latitude and longitude.

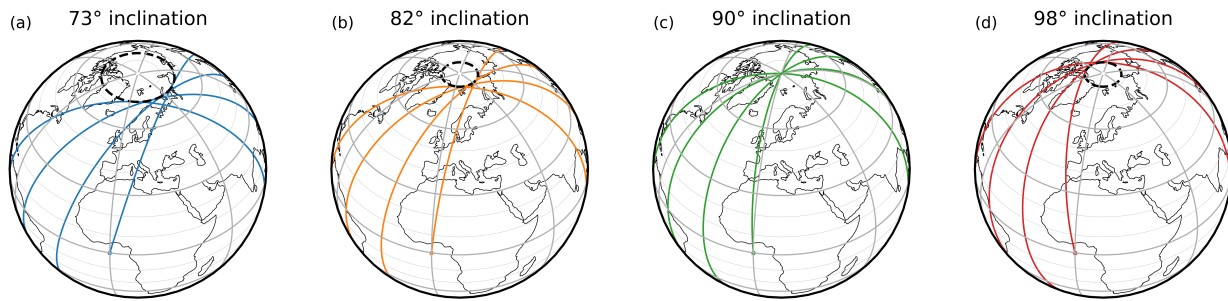

**Figure 5.** Satellite orbits with different inclinations. From (a) to (d): 73°, 82°, 90°(polar), 98°(sun-synchronous). The inclination describes the angle the satellite makes against the equator as it crosses from the southern hemisphere to the northern hemisphere, which is the same as the maximum latitude it reaches before returning south. Reproduced from Hocking et al. (2024).





In Sect. 3, we will present results for ensembles of single satellites and ensembles of satellite constellations. The individual members of each ensemble differ by initial RAAN, or equivalently by local solar time. The intention behind the choice of constellations is to consider combinations that can effectively mitigate the limitations of single satellites. For the constellations that use satellites with the same inclination, we only consider combinations with equal spacing in RAAN, in order to achieve balanced sampling of the diurnal cycle. In the case of a two-satellite constellation, for example, the two orbital planes are orthogonal to each other, and the two satellites thus observe local solar times 6 h apart. By contrast, constellations that use satellites with different inclinations may have any combination of initial RAAN values, since relative precession ensures that the difference in observed local solar time gradually changes over time. In this case, an ensemble for a two-satellite constellation has $12 \times 12 = 144$ ensemble members.

## 2.4 Synthetic wide-field-of-view measurements

We compute synthetic measurements at each time step, one minute apart. Our investigation considers the use of wide-field-of-view radiometers, where each measurement can be described as the result of the integral of radiances over all visible surface elements. Similarly to the measurement kernel used in Hocking et al. (2024), we express this integral over surface area rather than the conventional integral over solid angle (e.g. Gristey et al., 2017). In this work, we also include an anisotropy factor $R$, taken from the ERBE ADM (Sect. 2.2.3). The resulting integral is

$$F = \int_A dA \, R(\theta_0, \theta, \phi) \frac{\cos(\theta)\cos(\eta)}{\pi d^2} M, \tag{3}$$

where the measured quantity $F$ is the flux across the orbital surface, $A$ is the area of visible surface elements, $dA$ is an infinitesimal surface element, $\theta_0$ is the solar zenith angle, $\theta$ is the viewing zenith angle, $\phi$ is the relative azimuth angle, $\eta$ is the satellite view angle, $d$ is the distance between the satellite and the surface element, and $M$ is the radiant exitance of the surface element. We assume that the measurement instrument has a perfect cosine response ($\cos(\eta)$), and that there is zero pointing error. For our satellites, at an altitude of 700 km, the full instrument footprint has a diameter of close to 6000 km at the Earth's surface, but the measurement is dominated by the radiation from the central part of the footprint (Fig. 6).

## 2.5 Computation of the annual global mean

We use the same bin method as in Hocking et al. (2024), where we collect and process measurements in coarse 5°×5° latitude-longitude bins for each calendar year before computing the resulting global annual means. The processing sequence is illustrated schematically in Fig. 7. The method includes a simple, optional shortwave correction, to adjust the magnitude of shortwave measurements on an annual basis. This is to account for the fact that different satellites may sample the same parts of the diurnal cycle at different times of the year, when the overall magnitude of the incident solar radiation is different. Results for constellations of multiple satellites are normally computed with equal weights between constellation members. The exception is combinations of one 73° or 82° satellite with one 90° satellite, in which case the 90° data are only used to fill in the missing data near the poles.



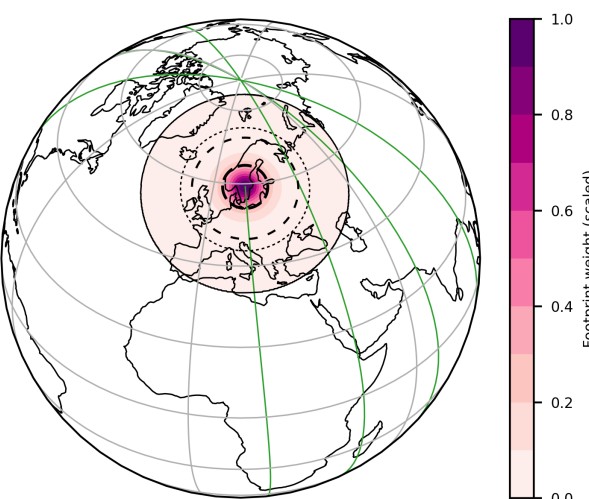

**Figure 6.** Satellite path and sample footprint. The solid green line shows the satellite path of a polar satellite. The solid black line shows the edge of the satellite footprint for a satellite at an altitude of 700 km. The footprint weights are shown for Lambertian radiation, and are scaled by a common factor such that the maximum value (immediately below the satellite) is unity. The dashed lines show the 50%, 90% and 95% thresholds for the cumulative response function from the centre of the footprint. Reproduced from Hocking et al. (2024).

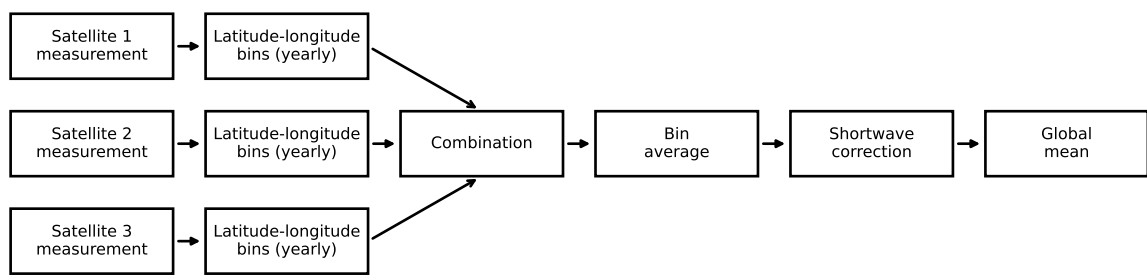

**Figure 7.** Schematic overview of the sequence of processing steps to convert satellite measurements into global mean values. Adapted from Hocking et al. (2024).

## 2.6 Comparison with the ideal angular distribution

Ideally, we would like to perfectly sample all combinations of viewing zenith angle, relative azimuth angle and solar zenith angle for each surface element, as if there were satellites everywhere around the Earth simultaneously. In practice, each single satellite will only see a limited number of surface elements at each time step, with only one combination of measurement angles



for each surface element. Nevertheless, the distribution of angular observations over time may be close to the ideal distribution. For a quantitative assessment, we then need to determine the shape of the ideal distribution, and quantify the relative similarity of the observed distribution to the reference distribution.

In the ideal reference distribution, the viewing zenith angle, the relative azimuth angle and the solar zenith angle are independent components. While all three may range from 0° to 180°, the only zenith angles we consider are between 0° and 90°, i.e. when the Sun and the satellite are above the horizon. For our analysis, we divided the distribution into discrete 10° bins for each of the three angles.

– The distribution of the viewing zenith angle is the result of the non-linear variation in zenith angle as a satellite passes from directly overhead to the horizon, weighted by the area of the spherical cap of the orbital sphere (Fig. 8a). Overall, larger zenith angles are more common than smaller zenith angles.

– The relative azimuth angle follows a flat distribution, by symmetry (Fig. 8b).

– The solar zenith angle depends on the latitude of the surface element, with a diurnal as well as a seasonal variation, such as when the polar regions experience polar night (Fig. 8c). We compute this distribution based on the actual solar zenith angle at each measurement time step.

The combined reference distribution in the resulting three-dimensional parameter space is then the product of each of the one-dimensional distributions, but this is not necessarily true for the observed distributions because their three angular components are not independent. Sample one-dimensional observed distributions are shown relative to the corresponding reference distributions in Figure 8.

As a simple method to assess the similarity of two distributions, we choose the overlapping coefficient (Inman and Bradley, 1989). Mathematically, the overlapping coefficient $OVL$ for two probability functions $f$ and $g$ can be defined as

$$OVL = \int_R \min(f(x),g(x))dx, \tag{4}$$

where $dx$ is an infinitesimal element in $R$, the domain of the functions. In one dimension, it represents the common area under the curves of the two probability density functions, as illustrated in Figure 8. In three dimensions, the corresponding quantity is the common hypervolume. Regardless of the number of dimensions, the result is a fraction between 0 and 1, where 1 indicates a perfect overlap between the two distributions. We present these fractions both as zonal profiles and as global means, and both for one-dimensional and three-dimensional distributions.

In terms of the interpretation of the overlapping coefficient, some caution is required when attempting to draw quantitative conclusions. To begin with, a perfect overlap is strictly speaking not sufficient to recover the correct radiation value. This is partly because the overlap is computed over longer time scales than the variations in the radiation field, and partly because different radiative situations may correspond to the same set of coordinates in the angular space. For example, the diurnal cycle of cloud cover and hence of reflectivity is asymmetric around local noon (Wylie, 2008), and the seasonal cycle of snow cover in the Northern Hemisphere is asymmetric around the December solstice (Robock, 1980), while the solar zenith angle is





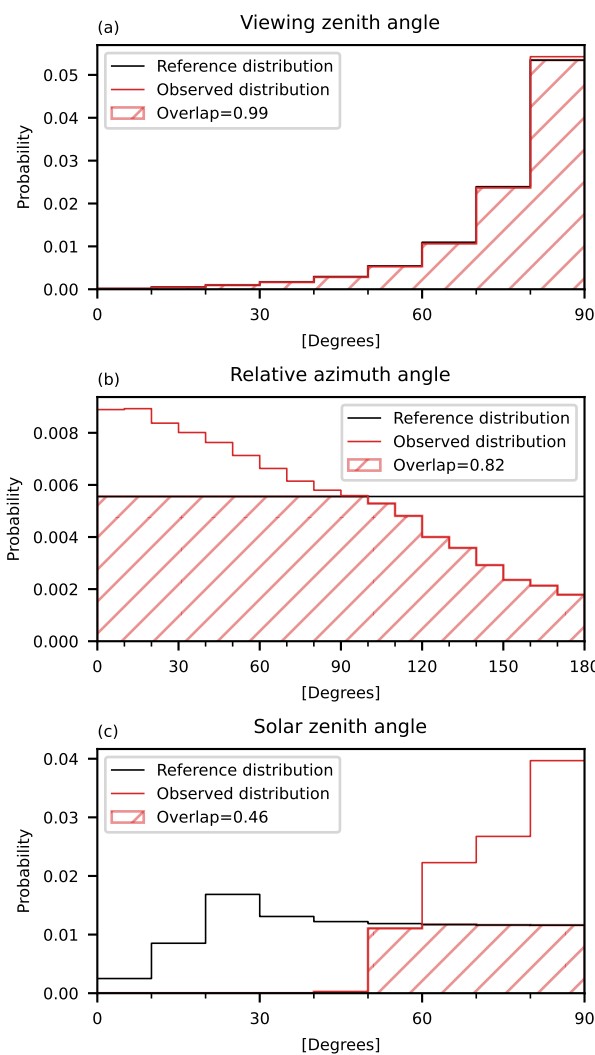

**Figure 8.** Sample angular distributions over one year for observations of a single surface element at the equator. The observed distributions are for a satellite in a sun-synchronous orbit at a local time of approximately 04:30 and 16:30. The hatched regions illustrate the overlap between the observed and reference distributions.

symmetric around noon and around the solstice. Furthermore, a single fraction does not inherently correspond to a quantitative estimate of deviation in global mean radiation, but a collection of these fractions make it possible to illustrate and to a certain extent quantify the relative angular sampling characteristics of different satellites. Finally, the combined three-dimensional overlap is not generally as simple as the direct product of the one-dimensional overlaps, but decomposition into separate
one-dimensional overlapping coefficients can still provide insight into the angular sampling behaviour of a given satellite.





## 3 Results

In this section, we present the main results of our study. First, we show our computed annual errors in global outgoing radiation relative to the reference truth, and how they change between a Lambertian assumption and our implementation of anisotropic shortwave reflection. This is followed by an investigation of angular overlap and the sampling characteristics that are associated
with the choice of satellite orbit, as well as a section on the overall geographical distribution of reflected shortwave radiation, notably at the poles. Then we briefly compare our results with those from Gristey et al. (2017). Finally, as a complement to the annual differences, we assess the impact of anisotropic shortwave reflection on estimated 5-year trends in the EEI.

### 3.1 Annual differences in reflected outgoing radiation

As described in Sect. 2, we have in this study simulated the global annual mean radiation at the TOA in an idealised satellite
measurement framework. Using simulated ensembles of satellites, we have investigated the difference in satellite measurements between cases with anisotropic and Lambertian shortwave reflection. These results are relevant in an assessment of the potential for wide-field-of-view satellite measurements of the EEI.

The results are presented quantitatively in the form of the root mean square error (RMSE) in the annual global deviations (Tables 2 and 3). The RMSE is computed over all the annual global differences for each ensemble over the five years, e.g. for an
ensemble with 12 members the RMSE is calculated from $12 \times 5 = 60$ separate annual global differences. Overall, the results for the various ensembles indicate that the effect of anisotropic shortwave reflection is limited for wide-field-of-view radiometers in the investigated orbits. The difference in shortwave-corrected RMSE between Lambertian and anisotropic reflection is at most 0.03 $\mathrm{Wm}^{-2}$, and often less, for precessing satellites and constellations that include them. This also applies to constellations of polar satellites, but single satellites have larger deviations of the order of 0.4 $\mathrm{Wm}^{-2}$. Sun-synchronous satellites have even
larger deviations, with single satellites resulting in RMSE deviations of several $\mathrm{Wm}^{-2}$ and even constellations of multiple sun-synchronous satellites resulting in RMSE deviations that are much larger than for other constellations with the same number of satellites.

Before presenting the annual results in more detail, we note that the shortwave correction results in an overall improvement for single satellites and two-member constellations. We therefore choose to consistently mention the annual differences in
Sect. 3.1.1 and 3.1.2 as shortwave-corrected values, unless otherwise described. The correction itself is briefly discussed in Sect. 3.1.3.

### 3.1.1 Individual satellites

For individual satellites, the effect of the type of shortwave reflection shows qualitatively different behaviour between sun-synchronous satellites and others; the former exhibit a clear change as a result of the introduction of anisotropic reflection, while the latter indicate a much smaller difference.





| Satellite inclination | 73°+90° | 82°+90° | 73°+82° | 2x 90° | 3x 90° | 4x 90° | 73° | 82° | 90° |
|---|---|---|---|---|---|---|---|---|---|
| Ensemble size | 144 | 144 | 144 | 6 | 4 | 3 | 12 | 12 | 12 |
| Anisotropic SW reflection | | | | | | | | | |
| RMSE [no SW corr] $(\mathrm{Wm}^{-2})$ | 0.38 | 0.17 | 0.26* | 0.19 | 0.03 | 0.02 | 0.42* | 0.17* | 3.12 |
| **RMSE [SW corr]** $(\mathrm{Wm}^{-2})$ | **0.13** | **0.12** | **0.08*** | **0.13** | **0.08** | **0.08** | **0.12*** | **0.11*** | **1.27** |
| Lambertian SW reflection | | | | | | | | | |
| RMSE [no SW corr] $(\mathrm{Wm}^{-2})$ | 0.44 | 0.20 | 0.30* | 0.31 | 0.05 | 0.02 | 0.48* | 0.20* | 3.50 |
| **RMSE [SW corr]** $(\mathrm{Wm}^{-2})$ | **0.10** | **0.11** | **0.08*** | **0.12** | **0.07** | **0.08** | **0.10*** | **0.11*** | **0.87** |
| Outgoing LW | | | | | | | | | |
| RMSE $(\mathrm{Wm}^{-2})$ | 0.05 | 0.05 | 0.04* | 0.04 | 0.04 | 0.04 | 0.04* | 0.04* | 0.12 |
| Net sum: LW + Anisotropic SW reflection | | | | | | | | | |
| RMSE [no SW corr] $(\mathrm{Wm}^{-2})$ | 0.39 | 0.18 | 0.26* | 0.20 | 0.04 | 0.04 | 0.43* | 0.18* | 3.22 |
| **RMSE [SW corr]** $(\mathrm{Wm}^{-2})$ | **0.11** | **0.08** | **0.07*** | **0.09** | **0.04** | **0.04** | **0.12*** | **0.08*** | **1.16** |
| Net sum: LW + Lambertian SW reflection | | | | | | | | | |
| RMSE [no SW corr] $(\mathrm{Wm}^{-2})$ | 0.44 | 0.21 | 0.30* | 0.31 | 0.06 | 0.04 | 0.48* | 0.21* | 3.60 |
| **RMSE [SW corr]** $(\mathrm{Wm}^{-2})$ | **0.09** | **0.08** | **0.06*** | **0.09** | **0.04** | **0.04** | **0.09*** | **0.09*** | **0.76** |

**Table 2.** Annual root-mean-square error (RMSE) in outgoing TOA radiation, with and without shortwave correction (SW corr), relative to the reference truth. These errors are computed over all ensemble members for the years 2001-2005. The values are based on measurements with different constellations, as identified by the inclination of the satellites. The 73°+90° and 82°+90° constellations use one precessing satellite in combination with one polar satellite to fill in data for the poles. The results for individual satellites (73°, 82°, 90°) are included for comparison. *These results only cover the latitude range of the satellites, and thus do not include the full polar regions. The effect of missing polar data is discussed in Sect. 3.3.

The precessing 73° and 82° satellites have similar performance regardless of the type of reflection, with up to 0.03 $\mathrm{Wm}^{-2}$ difference in RMSE (Table 2). It is important to note that this result does not include data from beyond the latitude range of the satellite, the effect of which is discussed further in Sect. 3.3. The polar 90° satellites achieve a performance with RMSEs that differ by a few tenths of a $\mathrm{Wm}^{-2}$ between anisotropic reflection and Lambertian reflection.

The sun-synchronous 98° satellites are more sensitive to how the reflection is treated, with un-corrected RMSEs that differ by over 1 $\mathrm{Wm}^{-2}$ (Table 3). Like the 73° and 82° results, this sun-synchronous result would in principle also require some kind of adjustment due to missing polar data, but single-satellite errors of over 20 $\mathrm{Wm}^{-2}$ are in any case so large that a single 98° satellite is not a realistic candidate for a mission to measure EEI with the current method. Modelling of the diurnal cycle can act to reduce these errors, but not to the levels that are required.




| Satellite inclination | 2x 98° | 3x 98° | 4x 98° | 6x 98° | 98° |
|---|---|---|---|---|---|
| Ensemble size | 6 | 4 | 3 | 2 | 12 |
| Anisotropic SW reflection | | | | | |
| RMSE [no SW corr] ($\mathrm{Wm}^{-2}$) | 4.46* | 1.14* | 0.25* | 0.15* | 23.08* |
| **RMSE [SW corr]** ($\mathrm{Wm}^{-2}$) | **0.24\*** | **0.26\*** | **0.13\*** | **0.08\*** | **26.56\*** |
| Lambertian SW reflection | | | | | |
| RMSE [no SW corr] ($\mathrm{Wm}^{-2}$) | 5.62* | 1.75* | 0.51* | 0.13* | 24.21* |
| **RMSE [SW corr]** ($\mathrm{Wm}^{-2}$) | **1.15\*** | **0.63\*** | **0.22\*** | **0.07\*** | **21.66\*** |
| Outgoing LW | | | | | |
| RMSE ($\mathrm{Wm}^{-2}$) | 0.11* | 0.02* | 0.04* | 0.02* | 1.03* |
| Net sum: LW + Anisotropic SW reflection | | | | | |
| RMSE [no SW corr] ($\mathrm{Wm}^{-2}$) | 4.53* | 1.13* | 0.29* | 0.15* | 24.08* |
| **RMSE [SW corr]** ($\mathrm{Wm}^{-2}$) | **0.28\*** | **0.26\*** | **0.12\*** | **0.06\*** | **25.73\*** |
| Net sum: LW + Lambertian SW reflection | | | | | |
| RMSE [no SW corr] ($\mathrm{Wm}^{-2}$) | 5.69* | 1.75* | 0.54* | 0.14* | 25.20* |
| **RMSE [SW corr]** ($\mathrm{Wm}^{-2}$) | **1.23\*** | **0.63\*** | **0.25\*** | **0.07\*** | **20.80\*** |

**Table 3.** As Table 2 for sun-synchronous satellites (98°).

### 3.1.2 Constellations of satellites

In terms of the anisotropy-related behaviour, the constellation results are qualitatively similar to the corresponding single-satellite results; sun-synchronous constellations show noticeable differences relative to the Lambertian case, while results for the other constellations are much closer between the two cases.

The combinations of two or more 73°, 82° or 90° satellites show markedly improved performance overall, and only small differences between Lambertian and anisotropic reflection (Table 2). The 73°+90° constellation has the largest difference at 0.03 $\mathrm{Wm}^{-2}$ for the shortwave value, and 0.02 $\mathrm{Wm}^{-2}$ for the total outgoing radiation. Overall, our results show that anisotropic reflection does not have a major impact on the sampling error in the EEI for these constellations.

In the case of sun-synchronous satellites, the anisotropic reflection leads to noticeable reductions in the RMSE compared with Lambertian reflection, sometimes by more than half (Table 3). Despite this improvement, there may be little impact in practice since the current methods mean that a constellation of sun-synchronous satellites still has a larger RMSE than a constellation with the same number of precessing or polar satellites.





### 3.1.3 Shortwave-corrected annual differences

As a brief comment on the shortwave correction method itself, in the context of the type of shortwave reflection, we note that its performance varies depending on the inclination of the relevant satellites. The correction typically does not perform as well in the anisotropic case as in the Lambertian case, which indicates that there is room for improvement of the method.

The single 90° satellites are a case where the shortwave correction clearly performs worse under anisotropy (Table 2). It was shown earlier that single 90° satellites are sensitive to insertion time because their orbital periodicity matches the periodicity

of the EEI anomalies (Hocking et al., 2024). The performance of the correction under anisotropic conditions may be indicative of systematic issues with the correction itself, which become apparent due to these periodicities in combination with enhanced reflection under certain conditions.

For two-satellite constellations of 73°, 82° and 90° satellites, the shortwave correction performs reasonably well, in that it reduces the RMSE to approximately $0.1\,\mathrm{Wm^{-2}}$ or lower, with the anisotropic correction being only slightly worse than in the

335 Lambertian case. With additional constellation members, the correction has a reduced impact, or even gives a slightly worse result in the case of polar satellites. However, the additional members themselves lead to improvements regardless.

For single 98° satellites, the shortwave-corrected results differ by almost $5\,\mathrm{Wm^{-2}}$, much more than the $\sim 1\,\mathrm{Wm^{-2}}$ difference without correction (Table 3). By contrast, for constellations with up to four 98° satellites, the shortwave-corrected RMSE in outgoing radiation with anisotropic reflection is less than half of the RMSE with Lambertian reflection.

### 3.2 Angular sampling

Let us now consider why the global mean estimates from these wide-field-of-view measurements exhibit such small sensitivity to the anisotropic reflection. As previously mentioned in Sect. 2.2.2, the anisotropic reflection does not change the total magnitude of the radiation, only its angular distribution. Furthermore, the anisotropic and Lambertian cases use the same satellite orbits by construction, so any difference in performance should only be the result of deviations in effective angular sampling.

We find that the reason why many satellite orbits give so similar anisotropic and Lambertian results is that these orbits provide sufficient sampling of the viewing zenith angle and the relative azimuth angle, as we will present in more detail below.

The four investigated satellite inclinations are associated different sampling characteristics in terms of the resulting angular distributions. The minimum-maximum ranges of global mean overlapping coefficients for ensembles with different inclinations are shown in Fig. 9, and sample values for a single satellite are shown in Table 4 for comparison. To understand our annual

radiation measurements, we are most interested in the annual overlaps. Perhaps the most striking feature of Fig. 9 is the distinct separation between the sun-synchronous satellites and the other satellites regarding annual overlap in particularly the solar zenith angle and therefore also the combined overlap. This illustrates the sampling impact of always observing at the same local time, which is not unexpected. The other satellites have annual overlaps that are close to 1, which we expect to be indicative of good sampling of the anisotropic radiation. On monthly time scales, the ensemble ranges are less clearly

separated, but the precessing 73° and 82° satellites clearly perform better overall. This is to be expected, given that they sample the diurnal cycle and hence the angular parameter space more frequently.





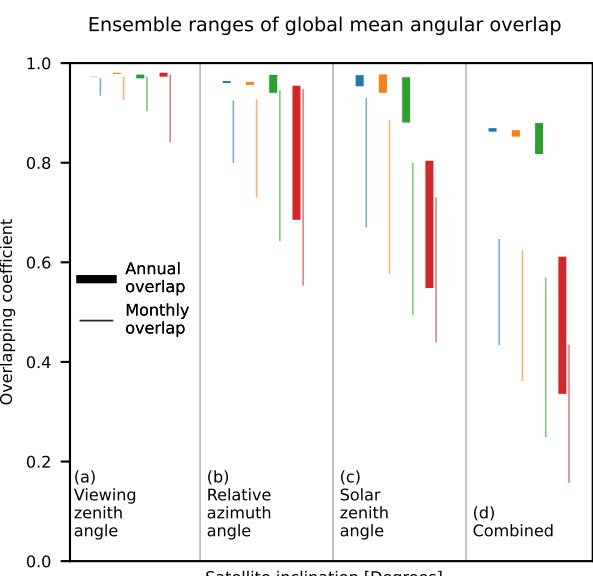

**Figure 9.** Global mean angular overlap for single satellites with various inclinations. This is a visualisation of the ensemble minimum-maximum ranges in Table 4, which also gives sample values for a single ensemble member.

| Inclination | 73° | (ensemble) | 82° | (ensemble) | 90° | (ensemble) | 98° | (ensemble) |
|---|---|---|---|---|---|---|---|---|
| Initial RAAN | 0° | 0°-165° | 0° | 0°-165° | 0° | 0°-165° | 0° | 0°-165° |
| Annual overlap | | | | | | | | |
| Viewing zenith angle | 0.97 | 0.97-0.97 | 0.98 | 0.98-0.98 | 0.98 | 0.97-0.98 | 0.98 | 0.97-0.98 |
| Relative azimuth angle | 0.96 | 0.96-0.96 | 0.96 | 0.96-0.96 | 0.98 | 0.94-0.98 | 0.79 | 0.69-0.95 |
| Solar zenith angle | 0.96 | 0.95-0.98 | 0.95 | 0.94-0.98 | 0.88 | 0.88-0.97 | 0.62 | 0.55-0.80 |
| Combined annual overlap | 0.86 | 0.86-0.87 | 0.86 | 0.85-0.87 | 0.82 | 0.82-0.88 | 0.45 | 0.34-0.61 |
| Monthly overlap | | | | | | | | |
| Viewing zenith angle | 0.96-0.97 | 0.93-0.97 | 0.96-0.97 | 0.93-0.97 | 0.91-0.97 | 0.90-0.97 | 0.92-0.97 | 0.84-0.98 |
| Relative azimuth angle | 0.88-0.92 | 0.80-0.92 | 0.80-0.91 | 0.73-0.93 | 0.72-0.94 | 0.64-0.95 | 0.69-0.80 | 0.55-0.95 |
| Solar zenith angle | 0.89-0.93 | 0.67-0.93 | 0.62-0.89 | 0.58-0.89 | 0.51-0.78 | 0.49-0.80 | 0.52-0.59 | 0.44-0.73 |
| Combined monthly overlap | 0.57-0.64 | 0.43-0.65 | 0.48-0.61 | 0.36-0.62 | 0.29-0.53 | 0.20-0.57 | 0.25-0.29 | 0.16-0.43 |

**Table 4.** Global average overlapping coefficients (Sect. 2.6) for individual satellites, computed over one calendar year. Sample values for a single satellite are indicated by the results for satellites initialised at 0° RAAN. For comparison, results are also given for an ensemble of 12 satellites, initialised at 0°-165° RAAN. All ranges are given as "minimum-maximum" ranges. The ensemble ranges are illustrated in Fig. 9. The zonal distributions for the 0° RAAN satellites are shown in Fig. 10.



**Figure 10.** Zonal distribution of the overlapping coefficient for the combined distribution of angles at a given surface latitude as observed by individual satellites with different orbital inclinations, compared with the ideal reference distribution. See Sect. 2.6 for more information about the method. All four satellites were initialised at 0° RAAN. Panels a, c, e and g show the range of monthly and annual overlaps as zonal profiles. Panels b, d, f and h show the variation in monthly overlaps over one year. The black regions mark polar night, during which it is not meaningful to compare the distribution of angles. Global mean overlapping coefficients for individual satellites, including those initialised with other RAAN values, are shown in Table 4.

As part of a more nuanced view of the geographical variations in the angular sampling, we also present zonal profiles of overlapping coefficients for four satellites in Fig. 10. On the annual time scale, the 73°, 82° and 90° satellites all result in observed angles that have a large overlap with the ideal distribution, and that have a high and almost constant overlap from the equator to approximately 80° latitude. By contrast, the sun-synchronous 98° satellite shows a lower overlap, which is especially clear in the tropics. However, it should be noted that the overlapping coefficient for the sun-synchronous satellites depends on the observed local time, and other sun-synchronous satellites have different characteristics (Table 4). In the monthly overlaps,




the differences are not quite as pronounced, but the 98° satellite still has a lower overlap in general and in the tropics in particular.

For all satellites, the regions closest to the poles show a clear decrease in the overlap compared with the rest of the profile. These regions are within the field of view of all four satellites, but the non-polar satellites do not pass directly overhead, and as such it is expected that they will miss some of the angular parameter space. By contrast, the 90° satellite does pass overhead. It is therefore interesting, and somewhat surprising, that the polar overlap from the 90° satellite is about the same as, or perhaps even worse than, the polar overlap from the 73° and 82° satellites. It may be that this apparent deficiency of the polar satellite is nevertheless compensated for by the fact that this satellite makes actual measurements all the way to the poles, which is important when computing the final mean value.

The difference between the Lambertian and the anisotropic case, as previously described, is that the latter takes into account the anisotropy through three angles: viewing zenith angle, relative azimuth angle and solar zenith angle. The Lambertian case already implicitly includes a certain dependence on the solar zenith angle, in that we use total magnitudes of reflected radiation at each surface element that are ultimately a result of the position of the Sun. In that case, good sampling of the solar zenith angle is necessary for good recovery of the annual mean. For the purpose of interpreting the differences in the global annual mean, the core distinction of the anisotropic case is then the variation with viewing zenith angle and relative azimuth angle. We conclude that the reason for the small differences in the annual means between anisotropic and Lambertian reflection, as presented in Sect. 3.1, is good sampling of the viewing zenith and relative azimuth angles.

## 3.3 Geographical distribution and polar brightness

The inclusion of anisotropy has a profound effect on the geographical distribution of reflected shortwave radiation as can be observed from space. Figure 11 shows the latitude profile of the difference between anisotropic and Lambertian observed outgoing reflected shortwave radiation. Overall, the poles appear brighter and the midlatitudes dimmer with anisotropic radiation than in the Lambertian assumption. This is likely due to large solar zenith angles at high latitudes combined with large viewing zenith angles and the shape of the ADMs for most scene types. For the 90° satellites, the change only results in a redistribution of the radiation, but the global mean remains the same. The 73° and 82° satellites, on the other hand, do not fully sample the bright poles, which results in a systematic deviation in the estimated global mean. It is interesting to note that the resulting latitude profiles are not only empty where the 73° and 82° satellites completely lack data, but the profiles start to deviate from the 90° profile already before the 73° and 82° satellites actually reach their maximum latitude. Given that the affected geographical area is small in these cases, the actual impact on the global mean is limited and may be possible to mitigate with external information, but the further the satellite inclination is from 90°, the larger this impact will be. If satellites at even lower inclinations are to be used to determine the global EEI, this systematic error near the poles needs to be accounted for.





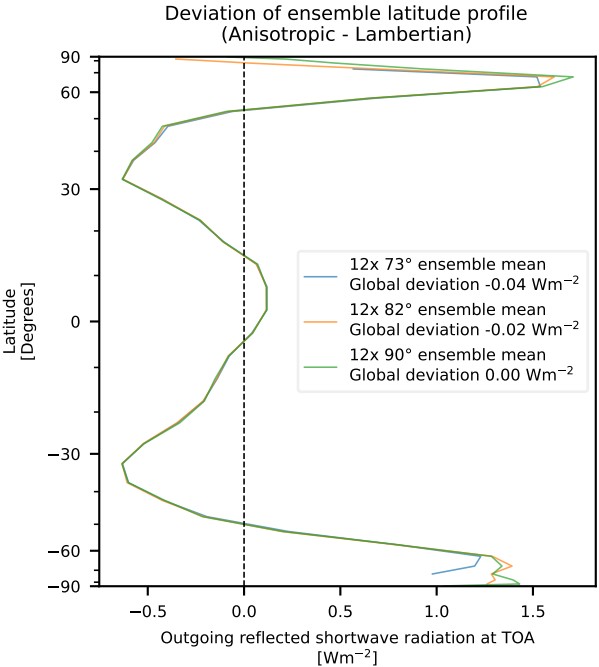

**Figure 11.** Mean latitude profiles of reflected shortwave radiation for ensembles of individual 73°, 82° and 90° satellites for the year 2001, showing the difference between anisotropic and Lambertian reflected radiation.

## 3.4 Comparison with Gristey et al. (2017)

The results presented in Sect. 3.1 indicate that there is only a limited difference in results between Lambertian and anisotropic
shortwave reflection, and that a constellation of satellites with wide-field-of-view radiometers can recover the annual global mean with a small error even when the shortwave reflection is anisotropic. At first glance, this would seem to contradict the results by Gristey et al. (2017), who found that the shortwave anisotropy led to a substantial bias of $0.56~\mathrm{Wm}^{-2}$ in the recovered global mean relative to a case with Lambertian reflection, over the course of their 24-hour simulation. However, further investigation suggests that the apparent discrepancy can be explained by differences in methodology.

A full replication of the Gristey et al. (2017) study is deemed outside the scope of the present study, which is to assess the impact on the annual global mean in the context of ECO: the proposed satellite mission to measure the global mean EEI on an annual time scale. Nevertheless, we make an effort to explore the apparent discrepancy. In order to produce more comparable results, we perform simulations for 36 satellites evenly distributed at an altitude of $780~\mathrm{km}$ and with an inclination of 86.4°, as in Gristey et al. (2017), but otherwise use our framework as previously described. We compute daily global mean values over
one year for the constellation, and find that the choice of Lambertian or anisotropic reflection occasionally leads to differences in the global mean of up to $0.5~\mathrm{Wm}^{-2}$ for single days, but the magnitude of the annual difference is less than $0.1~\mathrm{Wm}^{-2}$.



A similar analysis based on our ensembles of twelve $73°$ or $82°$ satellites shows daily differences of up to $0.7\ \mathrm{Wm}^{-2}$, but the annual difference remains less than $0.1\ \mathrm{Wm}^{-2}$ despite using fewer satellites.

The Gristey et al. (2017) difference of $0.56\ \mathrm{Wm}^{-2}$ is thus almost within the range of our largest single-day differences, and there are plausible explanations for the remaining discrepancy. In terms of temporal resolution, radiation data every 5 minutes, instead of hourly as in our study, are likely to capture more dramatic developments on short time scales. These fast developments in the reference data, in combination with local data gaps in the synthetic measurements on similar time scales, can lead to even larger differences between Lambertian and anisotropic reflection for individual days. Overall, we find it reasonable that anisotropic shortwave reflection could lead to a difference of $0.56\ \mathrm{Wm}^{-2}$ in the recovered global mean for a single day, while leading to a difference of less than $0.1\ \mathrm{Wm}^{-2}$ on an annual time scale. As such, what might seem like a different result compared with that from Gristey et al. (2017) can likely be explained by the differences in methodology between the two studies.

## 3.5 Trends

In addition to the absolute annual differences presented in Sect. 3.1, the effect of reflected shortwave anisotropy was also investigated in relation to estimated trends. Preliminary analysis of Lambertian data from Hocking et al. (2024), which covered a period of 20 years, indicated that the five-year trends were sensitive to the choice of five-year period. As such, it is not easy to determine an absolute five-year trend that would be representative for a longer time interval, and with the five years of data in the current study, we therefore chose to focus on the deviation in the estimated five-year trend as a result of anisotropic rather than Lambertian reflection. As shown in Fig. 12, the five-year trends showed little variation depending on the choice of shortwave reflection; the magnitude of trend differences between corresponding ensemble members was typically $\sim 0.01$ $\mathrm{Wm}^{-2}$ per decade, and never greater than $0.10\ \mathrm{Wm}^{-2}$ per decade. Our results from Hocking et al. (2024) showed that the sampling errors of wide-field-of-view measurements were low enough to allow EEI trend detection within 5 years, assuming Lambertian reflection; the trend results above indicate that this still applies even when more realistic shortwave reflection is included in the idealised framework.



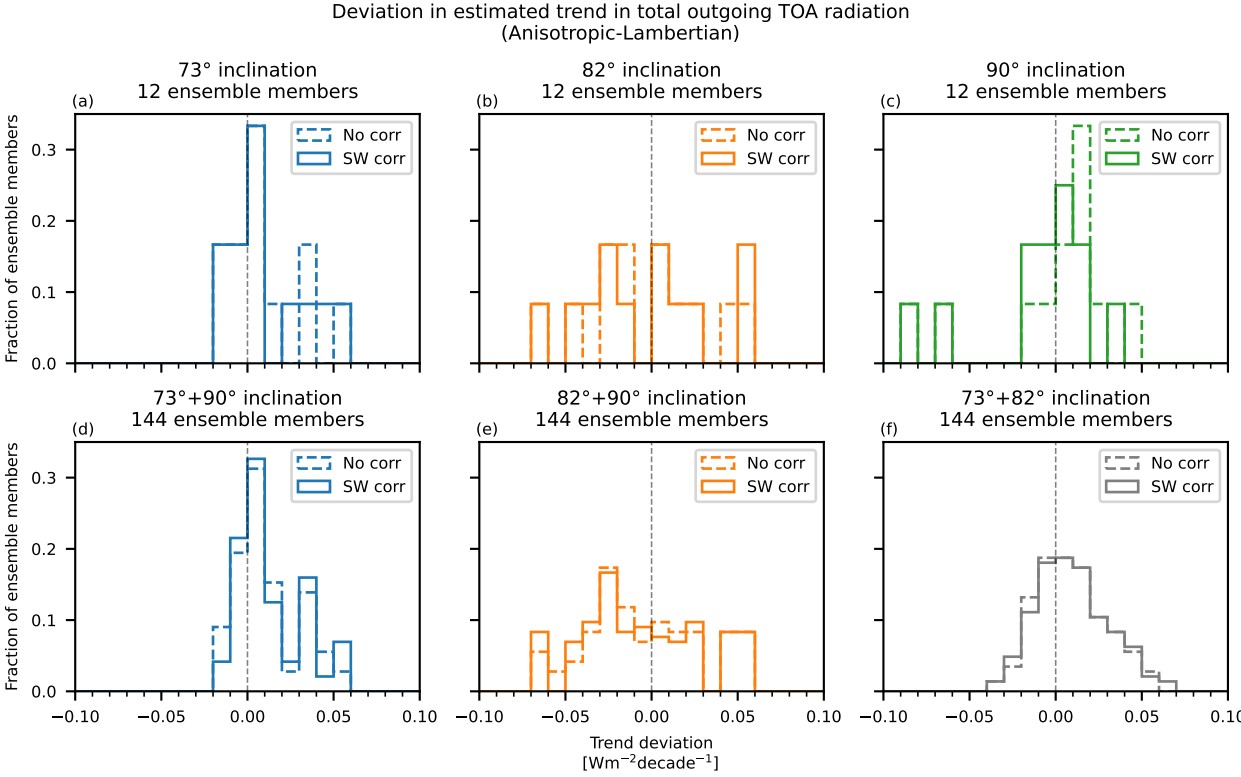

**Figure 12.** Difference in estimated five-year trends in outgoing total radiation between anisotropic and Lambertian reflection. Panels a-c show results for single satellites, while panels d-f show results for constellations of two satellites with different inclinations.. Each panel shows the distribution of results with and without shortwave-corrected measurements.



## 4    Conclusions

Monitoring of the EEI is fundamental to better understand our climate and allows us to predict future changes faster and more reliably. Current satellite missions gather highly valuable data on trends and absolute values of the EEI, but satellite-only estimates currently do not reach an absolute uncertainty below 1 $\mathrm{Wm}^{-2}$ in the annual global mean (Loeb et al., 2018). This absolute uncertainty is a target for future missions, which are designed with various measurement strategies in mind (Hakuba et al., 2019, 2024b; Hocking et al., 2024).

The proposed ECO mission considers wide-field-of-view radiometers, but previous research has indicated that shortwave anisotropy results in biased measurements of the global mean. In this study, we use angular distribution models in combination with top-of-atmosphere fluxes to compute radiances at hypothetical satellite locations and hence derive synthetic measurement time series for idealised wide-field-of-view radiometers. We investigate the effect of reflected shortwave anisotropy on the global annual mean as estimated by measurements from individual satellites or constellations of satellites in various orbits for a time period of five years. Our results provide additional context for indications of systematic biases from a previous study (Gristey et al., 2017), and we find that the shortwave anisotropy only has a small impact on the estimated values: at most 0.03 $\mathrm{Wm}^{-2}$ RMSE in the annual global mean, and typically only ∼0.01 $\mathrm{Wm}^{-2}$ per decade in the annual trend, when considering at least one precessing satellite.

We further investigate these results in terms of the underlying sampling of the angular parameter space, and find that the inclusion of anisotropic shortwave reflection makes it important to achieve good sampling of the viewing zenith angle and the relative azimuth angle, in addition to good sampling of the solar zenith angle that is already important without anisotropy (Hocking et al., 2024). The sun-synchronous 98° satellites have limited sampling of the solar zenith angle and the relative azimuth angle, which results in biased global mean measurements with our methods. By contrast, the precessing 73° and 82° satellites, and to some extent the 90° satellites, have a high overlap between their observed distributions and the ideal distribution on an annual basis. Even though high overlap is strictly speaking not sufficient for good recovery of the global annual mean, we argue that the small difference in results with anisotropic reflection for these satellites is explained by the high degree of overlap in viewing zenith angle and relative azimuth angle. This supports the idea that these satellites achieve adequate angular sampling on time scales of at least one year.

In terms of the spatial distribution of radiation, we find that anisotropic shortwave reflection results in dimmed midlatitudes and brighter poles relative to Lambertian reflection, which leads to a systematic bias in the global mean for non-polar satellites if the spatial redistribution is not accounted for. This bias is at most 0.04 $\mathrm{Wm}^{-2}$ for the inclinations studied here, but for inclinations even lower than 73° it will be increasingly important to account for this bias when determining the global mean.

Overall, our study shows that anisotropic shortwave reflection has only a limited impact on wide-field-of-view measurements of the EEI, provided that satellite orbits are chosen such that adequate angular sampling is ensured. As such, we conclude that wide-field-of-view satellite instruments have the potential to measure the EEI with small sampling errors, even when the shortwave reflection is anisotropic.



*Code and data availability.* The software used in this study will be published in the Bolin Centre Database. The ERBE shortwave ADMs were accessed from NASA Langley Research Center (2023) and are described in further detail in Suttles et al. (1988). The CERES SYN1deg

data are available from NASA/LARC/SD/ASDC (2017). The satellite orbits were computed with the simplified general perturbation model SGP4 (Vallado et al., 2006; Vallado and Crawford, 2008), as implemented in the Python library python-sgp4 (Rhodes, 2023). The computation of solar zenith angles relied on the Skyfield Python library (Rhodes, 2019).

*Author contributions.* This study was carried out by TH, who developed the software, developed the anisotropy implementation, performed the analysis and prepared the manuscript. All authors formulated the overarching goals of the study. MH initiated and developed the

470 anisotropy implementation, supported the analysis and contributed to the writing. LM and TM supported the analysis and contributed to the writing.

*Competing interests.* The authors declare that they have no competing interests.

*Acknowledgements.* This study benefitted from discussions with Jake Gristey, Dennis Hartmann, Seiji Kato, Norman Loeb, Roger Marchand and Robert Pincus. This project was funded by the European Research Council (ERC) (grant agreement no. 770765), the European Union's

Horizon 2020 research and innovation program (grant agreement nos. 820829 and 101003470), the Swedish Research Council (VR) (grant agreement no. 2022-03262), and the Swedish National Space Agency (grant agreement no. 2022-00108 and the ECO-e2e project). The computations were enabled by resources provided by the National Academic Infrastructure for Supercomputing in Sweden (NAISS) at the National Supercomputer Centre (NSC) partially funded by the Swedish Research Council through grant agreement no. 2022-06725. Contributing research by MH was carried out at the Jet Propulsion Laboratory, California Institute of Technology, under a contract with the

National Aeronautics and Space Administration (80NM0018D0004).



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
