# Peer review of "Impact of reflected shortwave anisotropy on satellite radiometer measurements of the Earth's energy imbalance"

_EGUsphere, 2025_

## Author Comment (AC1)

**Response to reviewers**

Thomas Hocking, Linda Megner, Maria Hakuba, Thorsten Mauritsen

**Introduction**

We are grateful to Seiji Kato and an anonymous referee for their comments on the manuscript "Impact of reflected shortwave anisotropy on satellite radiometer measurements of the Earth's energy imbalance" (egusphere-2025-829). The original comments are reproduced in black below, with our responses in blue.

A full record of the exact modifications in the revised document are available as a manuscript with tracked changes, uploaded separately.

**Comments by Anonymous Referee #1**

**Summary**

In their manuscript the authors study synthetic wide-field-of-view (WFOV) radiometer measurements onboard individual satellites or a constellation thereof to assess how accurately such measurements would quantify Earth's energy imbalance (EEI). To generate synthetic measurements, the authors use multiple years of the CERES SYN1deg product and mimic radiances by using either Lambertian or anisotropic scene reflection. The authors show that global EEI is largely independent of scene reflection, but strongly varies with type of orbit and multi-satellite constellation, owing angular sampling deficiencies of some orbits. Sorted by latitude, however, the authors present substantial EI differences. The manuscript is well-written and I recommend publication after minor revisions. As one aspect emerging from individual minor comments, I think the authors should add a discussion section.

We appreciate the suggestion for additional discussion of certain aspects. We have not added a specific discussion section because we prefer to keep these comments close to the corresponding sections in the text, but we have expanded our comments on certain topics, as detailed below.

**Minor points**

- l. 5 Please add "…sunlight and the observer." (or similar).
  Done

- ll. 5-7 The concept of assuming isotropic or anisotropic conditions has not been introduced, yet (and non-synthetic radiance measurements are not affected by such assumptions – only their subsequent L2 and L3 flux products). Consequently, this sentence sticks out and I recommend removing it.
  This sentence has been removed.

- ll. 10ff I recommend adding more specifics (e.g., which CERES product and ADMs used).
  Done

- Eq. 2: R is also a function of the scene (e.g., cloud cover, surface type). If possible, I would add the scene dependency after the angular dependency.
  The equation has been adjusted to show the scene dependency.

- l. 152 I would use the actual reference "(Su et al., 2015, Loeb et al., 2003) here and add "cloud microphysics (Tornow et al., 2021)".
  This has been rephrased to refer to "cloud properties" that are now elaborated at the end of the previous section. See also the next point.

- ll. 142-152 Similar to potential tendencies in cloud phase and optical depth, there is a chance that simpler ADMs miss anisotropic changes from cloud microphysics (e.g., as a result of fewer cloud condensation nuclei concentrations). I recommend adding this caveat in a designed discussion section (that is currently missing), as it may further impact flux deviations (shown in Fig. 12).
  We have expanded on this topic at the end of the previous section.

- Fig. 4 It is unclear whether WFOV was applied here or not. Please clarify.
  The caption has been modified to clarify that the figure uses synthetic WFOV measurements.

- Fig. 9 The thin lines are barely visible. Please make those lines thicker.
  This figure has been adjusted.

- Tab. 2 I fail to understand how "Ensemble size" is computed. Please improve the description in Section 2.3 and perhaps add another example.
  The description of ensemble size has been expanded, including an additional example.

- ll. 307-314 It is unclear to me why 98deg performs so poorly. Please discuss in a designated discussion section.
  We consider the poor performance of 98° with this method to be mainly related to the limited sampling of the diurnal cycle. The 98° performance was presented for Lambertian conditions in our previous publication (Hocking et al., 2024), and we feel that such a discussion would stray from the anisotropy focus of the current paper. We have added an explicit reference to the previous publication.

- ll. 381-392 I am surprised about the substantial regional differences and think it should be highlighted in the abstract. What are the implications for attribution (e.g., future radiative kernels or similar) that rely on accurate regional fluxes in combination with scene properties? Please discuss (ideally in a designated discussion section).
  It is challenging to compare satellite observations affected by anisotropy with model data calculated using plane-parallel geometry and isotropic fluxes. We note that this should not be a major issue with comparing models to the CERES data since the CERES products apply an angular dependence model. It is also planned to apply angular dependence models to the output of the cameras of the ECO mission, but the focus of the current paper is on the wide-field-of-view radiometers. We have added a brief comment in this section of the results, but a detailed discussion of the implications for attribution and feedback analysis is outside the scope of this paper. We now also mention the regional differences in the abstract.

- Fig. 12 The lines are hard to distinguish where there is overlap. I recommend adjusting line properties.
  This figure has been adjusted.

**Reference(s)** Tornow, F., C. Domenech, J. N. S. Cole, N. Madenach, and J. Fischer, 2021: Changes in TOA SW Fluxes over Marine Clouds When Estimated via Semiphysical Angular Distribution Models. J. Atmos. Oceanic Technol., 38, 669–684, https://doi.org/10.1175/JTECH-D-20-0107.1.

**Comments by Seiji Kato**

The authors investigate the error in top-of-atmosphere irradiances derived from a constellation of wide-field-of-view broadband instrument. The authors use ERBE angular distribution models and simulate wide-field-of-view instrument measurements. They consider isotropic radiation fields and radiation fields with anisotropy. They consider four different inclination angles. They express errors by RMS difference from reference truth data. They conclude that although anisotropy changes angular distribution of radiances, ignoring anisotropy does not introduce a significant error in derived irradiances.

I see a serious flaw in the method treating anisotropy to simulate wide-field-of-view measurements. Specifically, Equation (3) is not correct. In order to simulate wide-field-of-view measurements, the authors need integrate radiances over the field-of-view of the instrument weighted by cosign of the viewing angle. Integrating over the surface area does not include angular dependent of radiation fields properly. This is probably the main reason that the authors find that derived irradiances are insensitive to angular distribution of radiance. Please see a paper by Green et al. (1990) (Section 2) for correctly include angular distribution of radiances within a field-of-view of a wide-field-of-view instrument. Using the approach described in Green et al. the method to simulate wide-field-of-view observation using the SYN1deg-hour product is 1) estimate 1deg by 1deg grid boxes fall withing a field-of-view for a given sub-satellite point (i.e. one footprint), 2) obtain a scene type for each grid box within the field-of-view, 3) obtain viewing angle from the wide-field-of-view instrument to each grid box, 4) obtain solar zenith angle and azimuth angle for each glid box, 5) using the information, use a proper ERBE ADM and TOA flux from SYN1deg to infer the radiance toward the wide-field-of-view instrument, and 6) integrate all radiances weighted by cosign of the viewing angle over the field-of-view of the instrument.

Using ERBE AMDs to infer radiance in 6) above, you need to scale the SYN TOA irradiance by the ratio of (ADM radiance / ADM irradiance).

We agree that Green et al. (1990) present a valid approach to simulate wide-field-of-view measurements, and have clarified our notation and presentation to better explain our equivalent method. The integral shown in the original manuscript was intended to be close to the actual discrete integration that is done over the visible surface. We have now added a derivation.

The integral for the outgoing flux $F$ can be formulated fairly intuitively in terms of the radiance $L$, instrument view angle $\eta$ and instrument azimuth angle $\tau$. For a conical field of view, with maximum view angle $\eta_{max}$ and assuming a perfect $\cos \eta$ instrument response (adapted from Green et al., 1990):

$$F = \int_{\tau=0}^{2\pi} \int_{\eta=0}^{\eta_{max}} L(\eta, \tau) \cos \eta \sin \eta \ d\eta \ d\tau = \int_{\Omega_{FOV}} L(\Omega) \cos \eta \ d\Omega, \tag{1}$$

where $\Omega_{FOV}$ is the surface of visible solid angles within the instrument field of view.
The change of integration variable from solid angle to surface area uses

$$d\Omega = \frac{\cos \theta \ dA}{d^2}, \tag{2}$$

where $dA$ is the area of the infinitesimal visible surface element, $\theta$ is the satellite zenith angle, and $d$ is the distance between the satellite and the surface element. The integral for $F$ can then be expressed as:

$$F = \int_{A_{FOV}} L(A) \cos \eta \frac{\cos \theta}{d^2} \ dA = \int_{A_{FOV}} \frac{R(\theta_0, \theta, \phi) M}{\pi} \cos \eta \frac{\cos \theta}{d^2} \ dA, \tag{3}$$

where $A_{FOV}$ is the domain of visible surface elements within the field of view, $\theta_0$ is the solar zenith angle, $\phi$ is the relative azimuth angle, and $M$ is the radiant exitance of the surface element.
We account for each item of the step-by-step method as follows:

1. The 1° grid boxes that fall within one footprint are given by the domain $A_{FOV}$ of the integral.

2. The scene type is determined based on the cloud fraction and surface type for each grid box, and is implicitly included in the anisotropic factor $R$.

3. $\eta$ is the instrument viewing angle. $\theta$ is the viewing zenith angle.

4. The solar zenith angle and relative azimuth angle are given by $\theta_0$ and $\phi$.

5. The radiance $L$ toward the wide-field-of-view instrument is determined from the CERES SYN TOA flux $M$ and the anisotropy factor $R$ from the ERBE ADM:

$$L = \frac{R\,M}{\pi},\tag{4}$$

where the correct anisotropy factor is determined according to the SYN cloud fraction and ERBE surface type, in addition to the solar zenith angle, viewing zenith angle and relative azimuth angle.

6. After the change of variable to surface area, the integration over the instrument field of view is given by:

$$\int_{A_{FOV}} \frac{\cos\theta}{d^2} dA\tag{5}$$

The radiances are integrated over the instrument field of view, thanks to the domain $A_{FOV}$, with the weight given by $\cos\eta$.

Because we use input data on a discrete grid, the actual implementation is a sum:

$$F = \sum_i R_i(\theta_{0i}, \theta_i, \phi_i) \frac{\cos(\theta_i)\cos(\eta_i)}{\pi d_i^2} M_i A_i,\tag{6}$$

where the index $i$ indicates visible surface elements within the field of view, and the angles and the distance are computed from the centre of each surface element. We have added the intuitive formulation in equation 1 as a new starting point to the manuscript, and derived and clarified our discrete implementation of it.

These steps give one measurement of the wide-field-of-view and repeat these steps for all wide-field-of-view instrument footprints. Convert these wide-field-of-view measurements to irradiances at a reference height, you need to integrate changing viewing angles. This is done by shape factors described in Green and Smith (1991), but since you have angular distribution of radiances, you can directly integrate radiances changing viewing angles. The conversion factor to a reference height depends on angular distribution of radiances, hence the conversion factors for isotropic versus anisotropic are different. Loeb et al. (2002) describes how to determine a reference height. You might be able to skip this reference height conversion step by defining the viewing angle at the surface (i.e. the angle measured from the zenith viewing the instrument by an observer at the surface) for the integration in 6). The result is not the exactly the same, but the differences are probably small. These steps are needed to understand the error caused by ignoring anisotropy of radiances.

We agree that a wide-field-of-view measurement at different altitudes is affected by the viewing angles, and that e.g. a shape-factor method would be required to translate a measurement made at one altitude into what would be measured at another altitude. However, because we specifically consider the global mean values, these can be translated to equivalent global mean values at an alternative control altitude by a straightforward $(radius_1/radius_2)^2$ conversion factor thanks to conservation of energy. We do also define the zenith angles at the emitting surface, which we have placed at zero altitude.

To avoid misunderstandings, we have adjusted the figure showing the latitude profile of Anisotropic-Lambertian deviation (Fig. 11 in the original submission) to clarify that the profile is the satellite-level profile with magnitudes scaled to surface-level magnitudes.

**Minor comments**

I found the description of the method especially the section describing emulating shortwave reflection hard to understand. For instance, the authors use, radiance, irradiance, and radiation in the section. I do not understand what radiation means, it could be either radiance or irradiance. In addition, adding a figure helps explaining angular geometry.

We have adjusted our use of radiance, irradiance and radiation, and also clarified certain parts of the section "Emulating shortwave reflection". Figures have been added to illustrate the angular geometry of Sun-satellite relative positions and of radiance contributions to a wide-field-of-view measurement.

Lambertian is used to refer an isotropic field. I reserve the use of Lambertian for a perfect surface of which reflectance follows a cosign law.

The usage of "Lambertian" has been adjusted to more clearly refer to this kind of surface and the associated reflectance and radiance, rather than to an isotropic field.

References

Green R. N. and coauthors, 1990: Intercomparison of scanner and nonscanner measurements for the Earth Radiation Budget Experiment, J.Geophys. Res.95, 11785-11798.

Green, R. N., and G. L. Smith, 1991: Shortwave shape factor inversion of Earth Radiation Budget observations, J. Atmos. Sci, 48, 390-402

Loeb, N. G., S. Kato, B. A. Wielicki, 2002: Defining top-of-atmosphere flux reference level for Earth radiation budget studies, J. Climate, 15, 3301-3309.